

**Ionospheric Total Electron Content responses to HILDCAAs intervals**
Regia Pereira Silva[*1,2], Clezio Marcos Denardini[1], Manilo Soares Marques[3], Laysa
Cristina Araújo Resende[1,4], Juliano Moro[4,5], Giorgio Arlan da Silva Picanço[1], Gilvan
Luiz Borba[3], Marcos Aurélio Ferreira dos Santos[2].
[1]National Institute for Space Research – INPE, São José dos Campos-SP, Brazil.
[2]Northeast Regional Center – CRN/INPE, Natal-RN, Brazil.
[3]Geophysics Department (DGEF), Federal University of Rio Grande do Norte
(UFRN), Natal-RN, Brazil.
[4]National Space Science Center, China Academy of Science, CAS, Beijing, China;
[5]Southern Regional Space Research Center – CRS/COCRE/INPE, Santa Maria-RS,
Brazil.
[*]Corresponding author: INPE/DIDAE, Av. dos Astronautas, 1758, Jd. da Granja,
12227-010, São José dos Campos, SP, Brazil; Tel.: +55 12 3208-7184; e-mail:
regia.pereira@inpe.br; regiapereira@gmail.com



**Abstract**
The High-Intensity Long-Duration and Continuous AE Activities (HILDCAA)
intervals are capable of causing a global disturbance in the terrestrial ionosphere.
However, the ionospheric storms' behavior due to these geomagnetic activity forms is
still not widely understood. In this study, we seek to comprise the HILDCAAs
disturbance time effects in the Total Electron Content (TEC) values with respect to
the quiet days' pattern analyzing local time and seasonal dependences, and the
influences of the solar wind velocity to a sample of ten intervals occurred in 2015 and
2016 years. The main results showed that the hourly distribution of the disturbance
TEC may vary substantially between one interval and another. Doing a comparative
to geomagnetic storms, while the positive ionospheric storms are more pronounced in
the winter, this season presents less geoeffectiveness or almost none to HILDCAA
intervals. It was find an equinoctial anomaly, since the equinoxes represent more
ionospheric TEC responses during HILDCAA intervals than the solstices. Regarding
to the solar wind velocities, although HILDCAA intervals are associated to High
Speed Streams, this association does not present a direct relation regards to TEC
disturbances in low and equatorial latitudes.





*Keywords*: HILDCAA, TEC, Equatorial Ionosphere



## 1. Introduction

As similar to geomagnetic storms, High-Intensity Long-Duration and Continuous AE Activities (HILDCAA) intervals can influence the ionosphere, leading to disturbances in the ionospheric F2-region. It is well known that these intervals can change the F2-region peak height being, generally, less intense than those observed during typical geomagnetic storm events (Sobral et al., 2006; Koga et al., 2011, Silva et al., 2017).

In fact, HILDCAAs are characterized by present some criteria: i) the AE index must reach an intensity peak greater than or equal to 1000 nT; ii) The AE index needs to be almost continuous and never drops below 200 nT for more than two hours at a time; iii) The event must have a duration of at least two days, and iv) The event occurred after the main phase of magnetic storms. However, the same physical process may occur whether one of the four criteria are not strictly followed (Tsurutani and Gonzalez, 1987; Tsurutani et al., 2004; Sobral et al., 2006, Tsurutani et al., 2006; Hajra et al., 2013, Silva et al., 2017). As the main feature is the high AE index levels, in this study we have considered drops below 200 nT for more than two hours as long as the AE index value returns in high activity for prolonged hours.

The electron density perturbation in the ionosphere during HILDCAA events is different from that one occurred during geomagnetic storms in the equatorial and low latitudes stations. Since the HILDCAA presents a weak/moderate geoeffectiveness when it compares to the other forms of space disturbances, it is expected that the ionosphere response presents a differential behavior.

The Total Electron Content (TEC) is an important ionospheric parameter to several studies and technologic applications. As HILDCAAs can cause F2-region peak alterations, it can be observed the enhancements/depletions in TEC profile. In fact, the TEC response to the geomagnetic storms is a well-known issue in the space



physics field (Lu et al., 2001; Kutiev et al., 2005; Mendillo, 2006; Maruyama and
Nakamura, 2007; Biqiang et al., 2007; de Siqueira et al., 2011). However, only few
studies about TEC pattern during HILDCAAs intervals have been found in the
literature.
Ionospheric storms are manifestations of space weather events, which are caused by
energy inputs in the upper atmosphere in the form of enhanced electric fields,
currents, and energetic particle precipitation (Buonsanto, 1999; Mendillo, 2006).
Usually, ionospheric storms are associated with ionosphere responses to geomagnetic
storm events. However, in a broader way, these responses happen due to
magnetospheric energy inputs to the Earth's upper atmosphere, and this can occur to
all kind of geomagnetic activity form. Park (1974) pointed that ionospheric storms
can be understood in terms of the superposed effects of many substorm. In view of
the foregoing and considering that the development of ionospheric storms during
HILDCAAs intervals has not been dealt with in depth, in the current study we have
focused the TEC pattern during this kind of event.
Recently, Verkhoglyadova et al. (2013) suggested that HILDCAAs associated with
High Speed Streams (HSS) can be one of the external driving TEC variabilities.
Indeed, the continuous energy injection and energetic particles precipitation into the
polar upper atmosphere during HILDCAA intervals could modify the dynamic and
chemical coupling process of the thermosphere-ionosphere system resulting in
changes in the electron density. These modifications, beyond to change the auroral
electron density, can be mapped to low latitudes involving electric fields
disturbances, as prompt penetration electric fields (PPEF) and disturbance dynamo
(DD) (Koga et al., 2011; Silva et al., 2017).





Therefore, in the current study we have focused the TEC pattern during HILDCAAs
intervals, taking account local time dependence, seasonal dependence and high/slow
speed streams influences in the equatorial and low latitude ionosphere. This paper is
structured as followed: in the next section we present the HILDCAA intervals chosen
to support this study as well as the GNSS receivers locations over the Brazilian
region. In section 3 we show the results and discussion of the analysis and the
conclusions are presented in the last section.

**2. Data and Methodology**
In this study was possible to construct an overall perception of the ionospheric storms
occurred during HILDCAA disturbance time intervals that affect the TEC values with
respect to the expected behavior for quiet days. The features studied are local time
and seasonal dependences, and solar wind velocity influences.
We have selected ten HILDCAA intervals occurred during the 2015 – 2016 period.
These intervals are listed in Table 1, where the two columns present the identification
and the data range of each interval. The geomagnetic indices and interplanetary data
used   to   classify   the   HILDCAA   events   were   obtained   from   OMNIWeb
(https://omniweb.gsfc.nasa.gov/ow.html). The Kp index data were obtained from the
World   Data   Center   for   Geomagnetism,   Kyoto,   Japan   (http://wdc.kugi.kyoto-
u.ac.jp/kp/index.html). In this work it was used the daily Kp sum value.
The TEC mean was initially processed by a program developed at the Institute for
Space Research, Boston College, USA (Krishna, 2017). The mean values of vertical
TEC (VTEC) were obtained from two Brazilian GNSS stations, São Luís (SL) (2,59
S; 44,21 W) and Cachoeira Paulista (CP) (22,68 S; 44,98 W), representing the station
closest to the equator and the low latitude station, respectively. The Rinex files used





in this study were obtained from Brazilian Network for Continuous Monitoring of the
GNSS-RBMC Systems (RBMC) (https://www.ibge.gov.br/en/geosciences/geodetic-
positioning/geodetic-networks/20079-brazilian-network-for-continuous-monitoring-
of-the-gnss-systems-2?=&t=o-que-e). Besides that, the TEC data during HILDCAA
events were analyzed and then compared with a set of three days average belonging
to a quiet period, in which it refers to the three days less disturbed (ΣKp <24) of the
month of the occurrence of each HILDCAA interval.
Figure 1 shows a map with the location of each GNSS station, which is represented
by a red triangle. The dashed line represents the magnetic equator. The TEC data
obtained during the HILDCAA intervals were analyzed and then compared to the
TEC data during the selected quiet days, resulting in dTEC (dTEC = TEC mean –
TEC quiet days). All the analyses done in this work took into account the dTEC
values.

**3. Results and Discussions**
In this section, we will present the ionospheric TEC responses observed during ten
HILDCAA intervals focusing on local time dependence and seasonal features and the
solar wind velocity influences.

3.1 Local time dependence
A common feature of ionospheric storms is to be associated with dependence on local
time, mainly when they are caused by geomagnetic storms (Titheridge and
Buonsanto, 1988; Pedatella et al., 2010). However, to the best of the authors'
knowledge, no study has been found analyzing this aspect when regarding HILDCAA
intervals.



Figures 2 and 3 show the mean dTEC hourly values related to all HILDCAA intervals
for São Luís and Cachoeira Paulista, respectively. Each panel represents a single
interval from the bottom (H01) to the top (H10). The x axis is given in the Universal
Time (LT = UT − 3) and the color scale represents the dTEC values in TEC units
(TECu).
Notice that the dTEC values have a greater magnitude for the low latitude GNSS
station to the detriment of the closer equatorial GNSS station. The minimum and
maximum values are, respectively, -16.00 TECu and 27.40 TECu to São Luís, and -
37.60 TECu and 48.80 TECu to Cachoeira Paulista. It was considered the same
minimum and maximum values occurred to all intervals, for each station. This fact
explains why some intervals appear too close to the quiet time pattern. We believed
that since the HILDCAA events has low/moderate geoeffectiveness it was not
expected high values of the dTEC.
The distribution of the dTEC effects hour-to-hour during HILDCAA intervals shows
a substantial variability from one event to another. Habarulema et al. (2013) found
that the negative storms effects are observed during geomagnetic storms recovery
phases that over equatorial latitudes. However, since HILDCAAs intervals are
characterized by a long continuous phase of Dst index recovery, this does not apply.
The HILDCAA intervals present the positive dTEC predominance. In a more
simplified definition, HILDCAA means an interval where there is always energy
injection (Søraas et al., 2004; Sandanger et al., 2005). Silva et al. (2017) observed that
during HILDCAA intervals it was seen the uplift of the equatorial F2 region peak
height, probably due to prompt penetration electric fields. One of the main
mechanisms of TEC enhancements is the rise of the ionosphere to higher altitudes
where the recombination rates are small. Besides that, our results are in agreement





with the results found by de Siqueira et al. (2017). They did a study comparing the
TEC responses between two magnetic storms and two HILDCAAs intervals
following by them, and found a great TEC variability pattern from one to another
event. Hereupon, it was not possible to find a response pattern to the HILDCAA
effects in the equatorial and low latitude TEC considering only the local time. There
is great variability, and it is important to consider the day-to-day ionospheric
variabilities as well as the separate effect of each electric fields disturbance
(PPEF/DD).
Comparing both stations, Cachoeira Paulista GNSS station presented higher values
both to positive as negative ionospheric storms. During the daytime hours, the latitude
is responsible for the different ionospheric responses due to the presence of
photoionization. This probably explains the dTEC higher sensibility to low latitude
station in detriment of the closer equatorial latitude station.
Analyzing the hourly behavior of each interval from Figures 2 and 3, we observed
more intensity in TEC disturbances, both for positive and negative storms, during
some specific intervals. This aspect led us to make a seasonal analysis, which will be
presented in the next section.

3.2 Seasonal Dependence
It is well known for geomagnetic storms that the influence of the season entails on
positive/negative ionospheric storms is more pronounced in winter/summer than in
equinox months (Matsushita, 1959; Prölss and Najita, 1975; Mendillo, 2006, among
others). However, has not yet been established whether the occurrence of HILDCAA
interval in different seasons can do different TEC disturbances.





In a recent study involving more than one hundred HILDCAA events, Hajra et al.
(2013) reported no seasonal dependence, in what regards to predominant occurrence
rate in any specific epoch of the year due to the solar cycle influences. They
announced the HILDCAAs may occur during any month and any year, with increases
in the numbers of events occurring during the solar cycle descending phase. In the
current study, it was considered as seasonal dependence feature the TEC disturbances
responses at HILDCAA intervals already classified in a seasonal way.  The years
2015 and 2016 years comprise the descending phase of the $24^{th}$ solar cycle, which
made it possible to catalog an expressive number of HILDCAAs events in a short
time. Among the ten intervals chosen for this study, we have separated eight ones to
represent the seasonal variability, being two events for each station, taking into
account the month of occurrence of each interval, and considering the seasons as they
occur in South Hemisphere. The intervals are distributed according to the Table 2.
Figure 4 shows the disturbed TEC according to the seasonal classification which the
blue and coral colors refer to São Luís and Cachoeira Paulista, respectively. The solid
lines show an estimate of the central tendency for all values, minute-to-minute, for all
days of the events belongs to the season, while the shaded area represents the
confidence interval for that estimate. While the positive storms are more pronounced
in the winter for geomagnetic storms, to HILDCAA intervals this season presents less
geoeffectiveness, or almost none. Our results show that the equinoxes represent more
ionospheric TEC responses during HILDCAA intervals than the solstices. Both
equatorial and low latitude stations present positive storms during the autumn, while
the spring presents a negative behavior, mainly. This equinoctial anomaly may be
originated from the equinoctial differences in neutral winds, thermospheric





composition, and electric fields. Additional studies are necessary to quantify how
each factor can play an important role in HILDCAA seasonal TEC disturbances.

3.3 Solar wind velocities analysis
During the solar cycle descending phase, polar coronal holes migrate to lower
latitudes emanating intense magnetic fields. When HSS from these low latitudinal
coronal holes interact with slow speed streams (SSS) a region called Corotating
Interaction Regions (CIR) is formed and it is well characterized by compressions of
the magnetic field and plasma.
There are considerable works whose show how HILDCAA is well associate with
HSS and CIRs (Tsurutani et al., 2006; Verkhoglyadova et al., 2013). However, to be
associated not necessarily means that the degree of geoeffectiveness is directly related
to high speeds.
Figure 5 shows the solar wind velocities ($V_{SW}$) during each HILDCAA interval. As
the Figure 4, the blue and coral colors refer to São Luís and Cachoeira Paulista,
respectively. The diameter of the bubble is related to the velocity. The results showed
great variability from one interval to another, even considering the intervals that
occurred in the same year. In our first analysis (not shown here) we did not find a
direct association or cross-correlation between the VSW magnitude and the dTEC in
the equatorial and low latitude GNSS stations. Kim (2007) indicated that HILDCAA
intervals can be accompanied by HSS as well as SSS. It is possible to see in our
results that the dTEC responses to some intervals present similar behavior to both
HSS and SSS (e.g. H03, H07 and H08). This means that HILDCAA intervals can
affect the ionospheric TEC, but not in a direct correlation.



**4. Conclusions**

For this work, the ionospheric TEC response to a sample of ten HILDCAA intervals has been studied. We have used two GNSS stations from RBMC network representing equatorial and low latitude locations. As HILDCAA can affect the equatorial ionospheric F2 region, some disturbed TEC from its quiet time pattern is found. Addressing how the ionospheric storms behave during the HILDCAA intervals is our main goal.

Summarizing, HILDCAAs geoeffectiveness in Earth is mainly associated with CIRs, for this reason, the HILDCAA occurrence is more recurrent in the solar cycle descending phase since CIRs play a major role during this phase. Their effects occur during magnetic reconnection due to association with southward z component of the interplanetary magnetic field and Alfvén waves present in it (Tsurutani et al., 2004). These long-lasting intervals are due to continuous injection of energy and precipitation of particles, which disturb the high latitude ionosphere. The mainly disturbs are changes in thermospheric neutral composition, temperature, winds and electric fields. Similar to geomagnetic storms, theses disturbs can be mapped to low and equatorial latitude and alter the quiet time ionosphere. However, generally, they are less intense because in one astronomical unit the CIRs are not fully developed. In this study we seek to understand the behavior of the ionospheric storm during HILDCAA intervals. The main results are highlighted below:

- The hourly distribution of the dTEC during HILDCAAs intervals may vary substantially between low and equatorial latitude. Probably, the photoionization associated with latitude is responsible for these variations;





• Despite the geomagnetic storms recovery phase presents negative ionospheric
storms, this pattern do not occur during HILDCAA intervals. There is great
variability from one interval to another, but, predominantly, occurs positive phase;
• Regarding seasonal features, while the positive storms are more pronounced in the
winter for geomagnetic storms, this season present less geoeffectiveness, or almost
none to HILDCAA intervals. The equinoxes represent more ionospheric responses
to HILDCAA intervals presenting positive/negative phase predominance during
the autumn/spring;
• A well-known HILDCAA feature is its association with HSS present in the solar
wind. However, this association does not present a direct relation regards to TEC
disturbances in low and equatorial latitudes.
To conclude, the upshot of this study is the possibility to understand how ionospheric
storms behave during some HILDCAA intervals and to contribute to improving the
discussions about this issue.




**Data availability**
The data used in this work are made publicly available on the following sites:
https://omniweb.gsfc.nasa.gov/ow.html , http://wdc.kugi.kyoto-u.ac.jp/kp/index.html,
and https://www.ibge.gov.br/en/geosciences/geodetic-positioning/geodetic-
networks/20079-brazilian-network-for-continuous-monitoring-of-the-gnss-systems-
2?=&t=o-que-e . The GPS-TEC program used in this work is available in
http://seemala.blogspot.com/

**Author contributions**
R. P. Silva conceived the study, designed the data analysis, discussed the results and
leaded writing this manuscript.
C. M. Denardini assisted to conceive the study, to design the GNSS data analysis and
discuss the final results.
M. S. Marques assisted with the GNSS data analysis and with designing the figures.
L. C. A. Resende assisted to design the study and discuss the results of the study.
J. Moro assisted to design the study and discuss the results of the study.
G. A. S. Picanço assisted to discuss the results of the study and review the
manuscript.
G. L. Borba assisted to discuss the results of the study and review the manuscript.
M. A. F. Santos assisted to discuss the results of the study and review the manuscript.
All the authors helped to write and to revise the manuscript.

**Competing interests**
The authors declare that they have no conflict of interest.





**Special issue statement**

This article is part of the special issue "7th Brazilian meeting on space geophysics and aeronomy". It is a result of the Brazilian meeting on Space Geophysics and Aeronomy, Santa Maria/RS, Brazil, 5–9 November 2018.

**Acknowledgements**

R. P. Silva acknowledges the supports from Conselho Nacional de Desenvolvimento Científico e Tecnológico (CNPq) through the Grant No. 300329/2019-9. C. M. Denardini thanks to CNPq/MCTIC (Grant 303643/2017-0). L. C. A. Resende would like to thank the National Space Science Center (NSSC), Chinese Academy of Sciences (CAS) for supporting her postdoctoral. J. Moro would like to acknowledge the China-Brazil Joint Laboratory for Space Weather (CBJLSW), National Space Science Center (NSSC), Chinese Academy of Sciences (CAS) for supporting his Postdoctoral fellowship, and the National Council for Scientific and Technological Development (CNPq) for the grant 429517/2018-01. G. A. S. Picanço thanks CAPES for supporting his Ph.D. (Grant 88887.351778/2019-00). We also would like to thank the OMNI data (https://omniweb.gsfc.nasa.gov/ow.html), and the World Data Center for Geomagnetism, Kyoto (http://wdc.kugi.kyoto-u.ac.jp/kp/index.html). The Rinex files were obtained from Brazilian Network for Continuous Monitoring of the GNSS-RBMC Systems (RBMC) at interface https://www.ibge.gov.br/en/geosciences/geodetic-positioning/geodetic-networks/20079-brazilian-network-for-continuous-monitoring-of-the-gnss-systems-2?=&t=o-que-e. The authors acknowledge Gopi Seemala for making available the GPS-TEC program (http://seemala.blogspot.com/).



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



**Figure captions**
**FIGURE 1** − Map showing the locations of the GNSS stations used in the present
study. Both stations are localized in the Brazilian region and are marked by a red triangle,
where SL and CP are, respectively, São Luís and Cachoeira Paulista.
**FIGURE 2** − dTEC hourly values to all HILDCAA intervals to São Luís (equatorial
station).
**FIGURE 3** − dTEC hourly values to all HILDCAA intervals to Cachoeira Paulista
(low latitude station).
**FIGURE 4** − Seasonal dTEC response to HILDCAA intervals. The blue and coral
lines refer to São Luís and Cachoeira Paulista, respectively.
**FIGURE 5** − Solar wind velocities analysis during HILDCAA intervals. The blue
and coral colors refer to São Luís and Cachoeira Paulista stations, respectively, while
the bubble diameter is related to velocity (km/s).




**Table captions**
**TABLE 1 –** The date range for HILDCAA intervals identified during 2015 – 2016
years
**TABLE 2 –** Seasonal classification of HILDCAA intervals (according to the seasons
in the Southern hemisphere).



**FIGURE 1** –

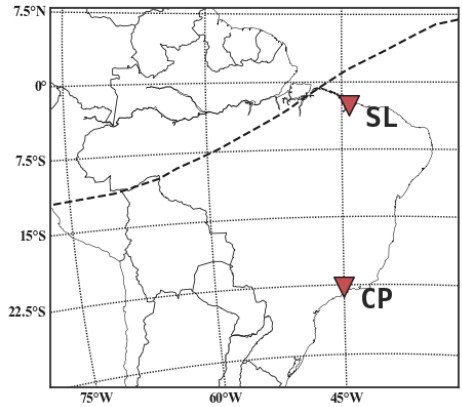






**FIGURE 2** -

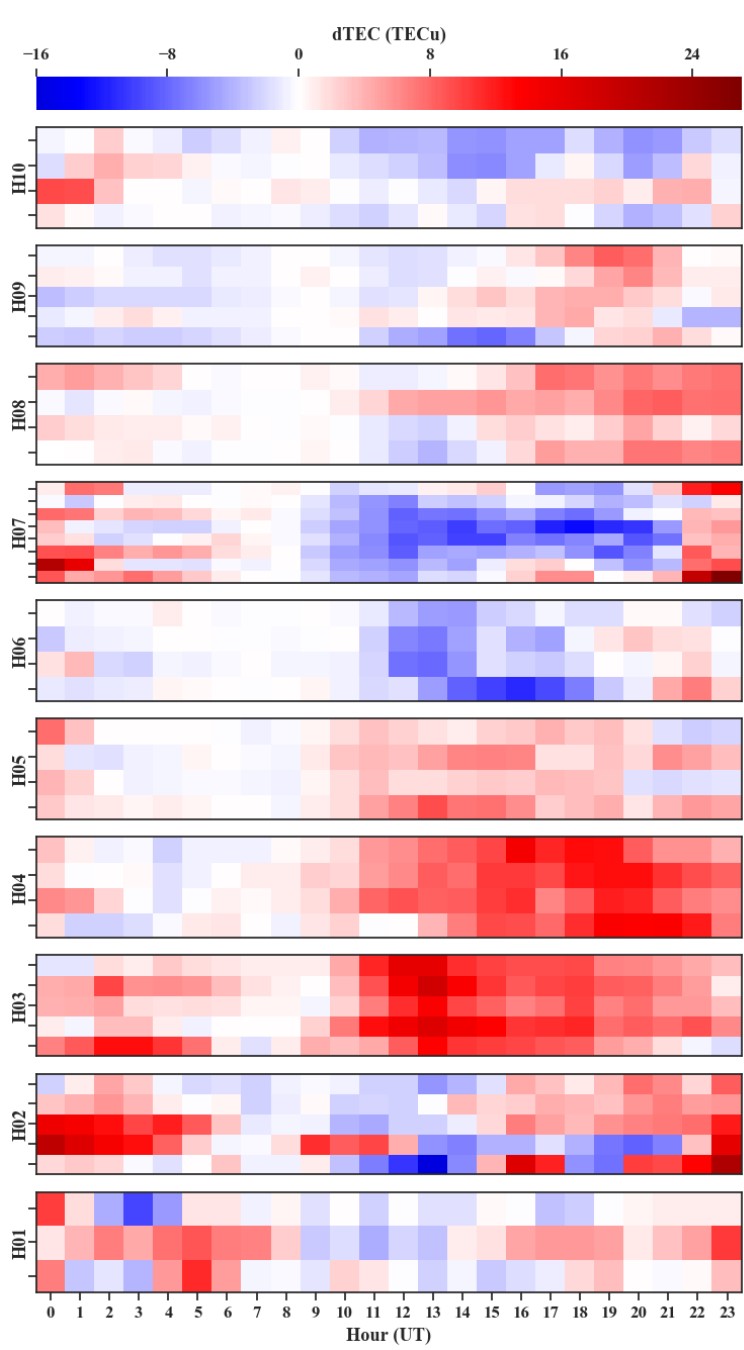








**455** **FIGURE 3** −

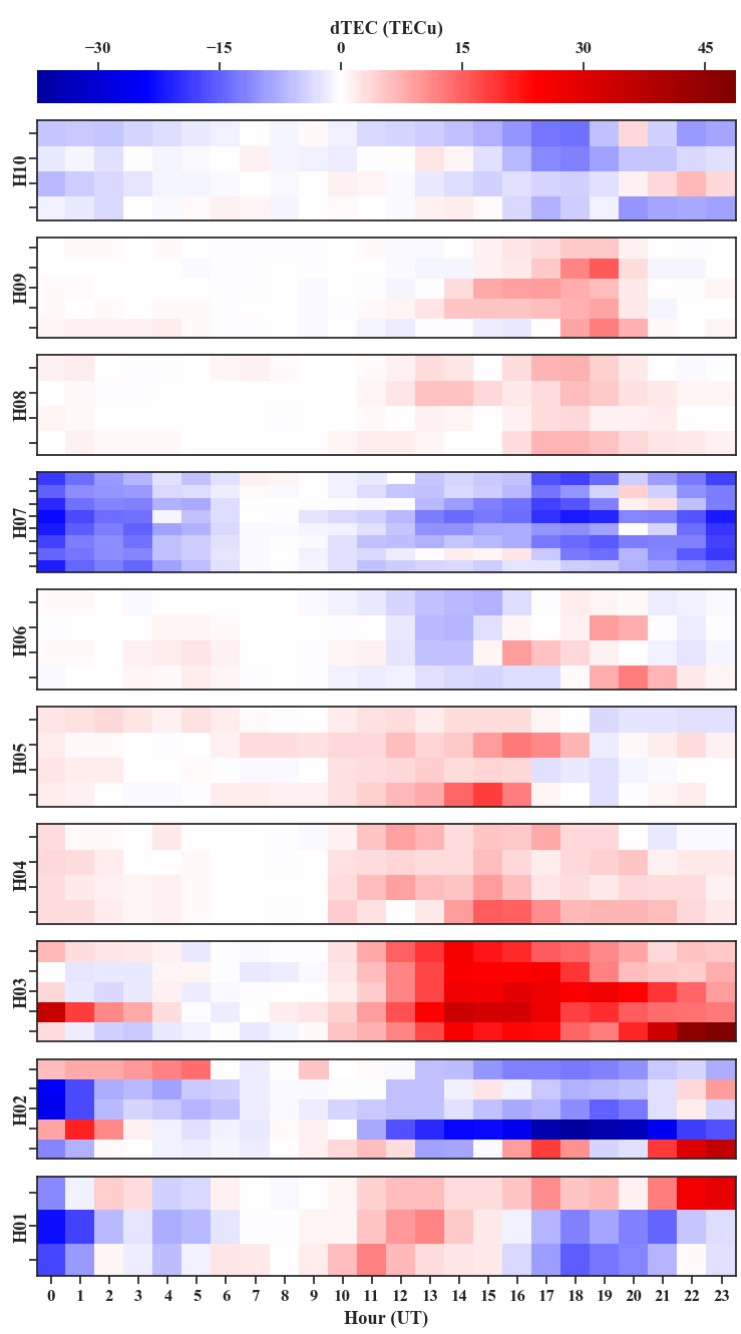






**FIGURE 4** −

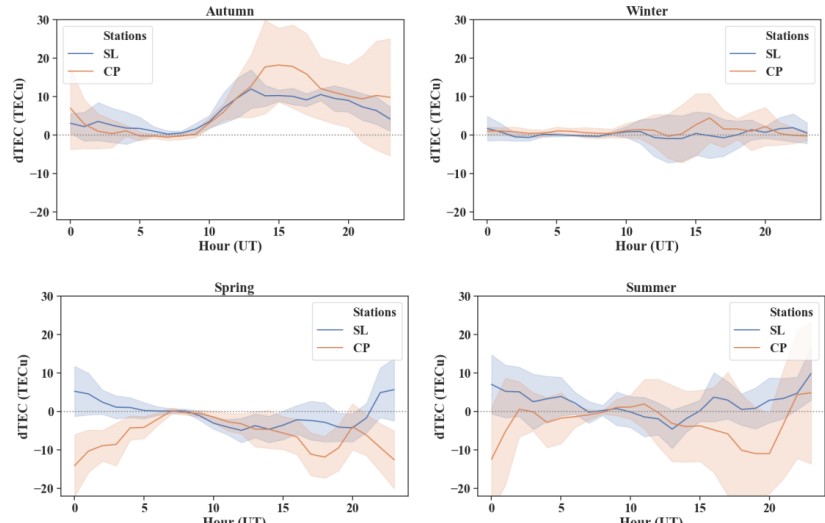







**FIGURE 5** −

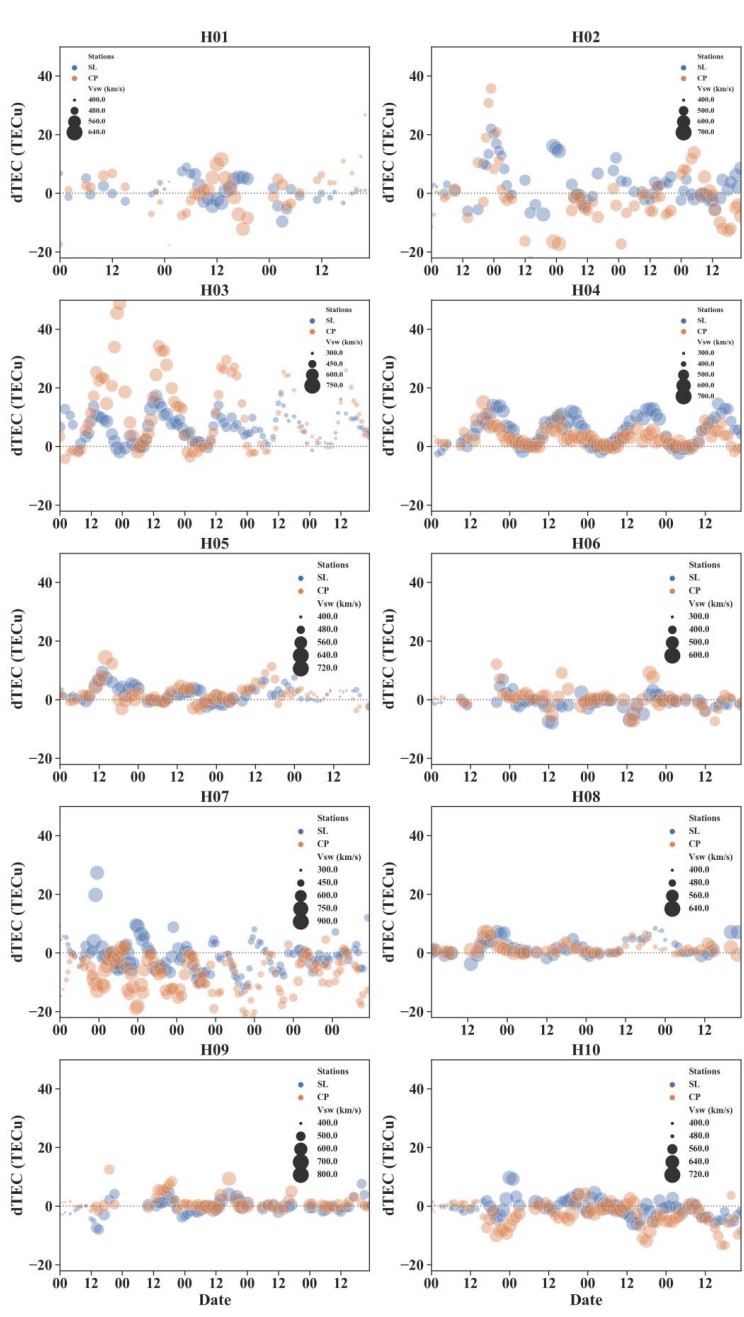







**TABLE 1** –

| ID | Date range |
|----|------------|
| H01 | 2015/03/01 – 03 |
| H02 | 2015/03/17 – 21 |
| H03 | 2015/04/16 – 20 |
| H04 | 2015/06/08 – 11 |
| H05 | 2015/07/11 – 14 |
| H06 | 2015/08/15 – 18 |
| H07 | 2015/10/07 – 14 |
| H08 | 2016/07/09 – 12 |
| H09 | 2016/08/03 – 07 |
| H10 | 2016/12/08 – 11 |








**TABLE 2** –

| Season | HILDCAA Intervals |
|--------|-------------------|
| Autumn | H03 and H04 |
| Winter | H05 and H06 |
| Spring | H07 and H10 |
| Summer | H01 and H02 |
