# Peer review of "Ionospheric Total Electron Content responses to HILDCAAs intervals"

_Annales Geophysicae, 2019_

## Referee Comment (RC1) · Thana Yeeram (Referee) · 3 Sep 2019

General comments: I have read the manuscript that reports the responses of TEC to HILDCAA intervals over the two Brazilian GNSS stations. The manuscript is fairly good in presentation, particularly in reporting an equinoctial anomaly of the TEC during HILDCAAs. As known, this topic has not been extensively studied and is in progress in the field. Since some of issues in the present forms are not adequately explained for the underlying Physics, I decided a minor revision for this manuscript.

Specific comment: 1. The abstract should be written in a concise form for the lines 32 - 39. 2. I think that one of the HILDCAAs' criteria is there are HSS and high frequency fluctuations of IMF Bz about zero value. 3. In lines 65 - 67, the authors may refer to

no 2. for HILDCAAs' properties as well. 4. In lines 74 - 75, the authors should mention the references. 5. In section Data and Methodology, the authors should remove the links of data since they are already in the acknowledgement. 6. In line 218 please give more details about the related mechanism for the equinoctial anomaly. 7. In section 3.3 why the solar wind speed is thought to be a main factor that affects the TEC? 8. I would like to suggest some related work that may fulfill the discussion: For recent study of TEC and HILDCAA: -de Siqueira Negreti, P. M., de Paula, E. R., and Candido, C. M. N.: Total electron content responses to HILDCAAs and geomagnetic storms over South America, Ann. Geophys., 35, 1309–1326, https://doi.org/10.5194/angeo-35-1309-2017, 2017.

for PPEF and DDEF during HILDCAA: -Yeeram, T. (2019). The solar wind - magnetospheric coupling and daytime disturbance electric fields in equatorial ionosphere during consecutive recurrent geomagnetic storms, Journal of Atmospheric and Solar-Terrestrial Physics, 187, 40-52.

-Yeeram, T., and Paratrasri, A. (2018). Recurrent geomagnetic storms and equinoctial ionospheric F-region in the low magnetic latitude: a case study, IOP Conf. Series: Journal of Physics: Conf. Series 1144: 012024(1-4).

9. Line 251, the geoeffectiveness of HILDCAA can be different and separated from CIR. Please see Hajra et al. 2015. Relativistic electron acceleration during HILDCAA events: are precursor CIR magnetic storms important? 10. The conclusions should be written in a short and concise form. 11. I do not understand the x-axis of Figs 2 - 5. Why the scale is for one day and for what? I see HILDCAA intervals are longer than that for each event H? Please describe in the text.

The technical comment 1. Some of the English must be corrected. For example, line 251, gerund "Summarizing" should be -> In summary/In conclusion. 2. The sentences must be simple and concise.

---

## Referee Comment (RC2) · Anonymous Referee #2 · 9 Sep 2019

This work deals with a topical issue, which is space weather and its effect on the ionosphere. However I cannot see which is the real message. If I am right, I understand that the authors want to show that HILDCAAs affect TEC at low and equatorial latitudes. But in my opinion, they need to show that what they find is not something at random, and to exlude any other source of disturbance. I think that the paper may become acceptable for publciation after some revision.

My main comments are the following:

1) I would add some statistical analysis, or a qualitative analyses, to show that the TEC disturbances they observe are due to HILDCAAs, by showing that the variation they see is not at random and is not due to any other mechanism. Is it possible that it is due to geomagnetic storms, for example? Maybe I am not fully understanding HILDCAAs.

[Figure]

2) Line 165: "The HILDCAA intervals present the positive dTEC predominance" Can you quantify this ?

3) In Figures 2 and 3: what is the "y" axis?

4) In Figure 5, for H03 and H04 a clear daily variation can be noticed. This 24-hour oscillation is absent in the others. What does this mean?

Minor comments: 1) Line 53: "The", should be lower-case, that is "the", and also in the following "The" which appear after a semi colon.

2) Line 67: I thinkg that may be "differential" should be "different"

3) Line 97: "taking account" should be "taking into account"

3) Line 118: use dot for decimal of latitude and longitude instead of comma. That is "2,59" should be 2.59

4) Line 153:Why "to São Luís". Shoudn't it be "for São Luís" or at Sao Luis? The same in the case of Cachoeira Paulista

5) Line 154: What do you mean by "It was considered the same minimum and maximum values occurred to all intervals, for each station." That you used the same scale range ? If this is the case, then you should explain it properly.

6) Line 162: "...phases that over equatorial latitudes", may be you mean "...phases over equatorial latitudes" ?

7) Line 228: "There are considerable works whose show" should be "There are considerable works that show ...". That is, that instead of whose.

---

## Author Comment (AC1) · 1 Oct 2019

Manuscript "Ionospheric Total Electron Content responses to HILDCAAs intervals" by Regia Pereira Silva et al. submitted to Annales Geophysicae.

We would like to thank the Referee for all the suggestions and comments given in order to improve this paper. We really hope that all doubts have been clarified.

Responses to specific comments: 1. We would like to thank the Reviewer for this suggestion. We modified the abstract in order to let it more concise. 2. Actually, the criteria of the fluctuation of IMF Bz around zero was added in a later study carried out by Koga et al. 2011 (Doi: 10.1016/j.jastp.2010.09.002). According to Tsusutani et al., 2004 (Doi: 10.1016/j.jastp.2003.08.015) the same physical process may occur

weather one criteria is not strictly followed. 3. We would like to thank the Referee for this suggestion. However, as the information written in the lines 65-67 is well-known, we have preferred to let it in this way. 4. Thank you for the suggestion. We added the reference. 5. Thanks for your suggestion. We removed the link of data from the Data and Methodology section and left it in the Acknowledgement section. 6. Information about the equinoctial anomaly is already written in the manuscript in lines 223 - 225. For more details, please see Balan et al., 1998 (doi:10.1029/97JA03137), Mansilla et al., 2005 (doi:10.1016/j.jastp.2005.02.024) and Chen et al., 2012 (doi:10.5194/angeo-30-613-2012) 7. We would like to thank the referee for this question. We realize that several papers address the relation between HILDCAAs and HSS/CIRs. However, how this relation is done is still an open question. Contrary to our expectations, the TEC answers during HILDCAAs are no direct relation to fast speeds. 8. We would like to thank the Referee for the suggestions. We read them and added the citation in lines 75, 95 and 237. 9. Yes, the geoeffectiveness of HILDCAAs can be separated from CIRs, since to a lesser extent, there are HILDCAA events related to CME. 10. We would like to thank the suggestion. The main results were written in topics in the Conclusions section. 11. The scale is not for one single day. The x-axis in Figures 2, 3 and 4 represent the mean dTEC hourly values. For Figures 2 and 3, the mean values were represented in this way in order to allow the comparison between one interval and other (please, see the lines 147 - 148). For Figure 4 the x-axis represents the central tendency for all dTEC values, minute-to-minute, as can be seen in lines 215 - 218. Lastly, in Figure 5, the x-axis represents the time duration of all intervals, represented each 12 hours.

Responses to technical comments: 1. We would like to thank the Referee for this correction. The word was changed in the manuscript. 2. We are very thankful to the Referee to help us improve the manuscript.

Please also note the supplement to this comment:
https://www.ann-geophys-discuss.net/angeo-2019-105/angeo-2019-105-AC1-

supplement.pdf

---

## Author Comment (AC2) · 1 Oct 2019

Manuscript "Ionospheric Total Electron Content responses to HILDCAAs intervals" by Regia Pereira Silva et al. submitted to Annales Geophysicae.

We are very grateful for all the useful suggestions and observations given by the Reviewer in order to improve this paper. Especially to the observations about typing errors in the manuscript. We really hope that all doubts have been clarified.

Responses to main comments: 1. Unfortunately, do a statistical analysis with ten HILDCAA events as a sample is not significant. However, we believe that carried out a qualitative analysis when we seek to comprise the HILDCAAs disturbances regards seasonal behavior or local time, for example. All TEC disturbances showed in the

study is about HILDCAAs influences, since all analysis was done taking account the dTEC (dTEC = Tec mean - TEC quiet days). Please, see lines 125 - 128. HILDCAAs are manifestations of space weather events in the form of geomagnetic activities. Their main feature is a continuous flux of energetic particles into the magnetosphere. The average Dst may remain suppressed for days or weeks, and appear as an unnaturally long storm recovery phase. For more, please see Tsurutani and Gonzalez, 1987 (doi: 10.1016/0032-0633(87)90097-3), Tsurutani et al., 2004 (doi: 10.1016/j.jastp.2003.08.015), Hajra et al., 2013 (doi: 10.1002/jgra.50530) and Silva et al., 2017 (doi: 10.5194/angeo-35-1165-2017). 2. We would like to thank the Referee for this suggestion. We realized that the sentence was short and no concise. To improve the understanding, we have adding the following sentences (lines 168 - 170): "60% (70%) of all intervals present a positive dTEC response during the whole event for São Luís (Cachoeira Paulista)." The percentage is regarding the predominance of the positive dTEC during the whole hourly distribution. Only four events for São Luís (three for Cachoeira Paulista) had negative predominance of the TEC behavior. 3. Figures 2 and 3 are the subplots of all HILDCAA intervals regarding mean dTEC hourly distribution, i. e., each panel refers to each interval. The y-axis is the HILDCAA interval according to its identification (H01, H02, etc). However, the y-axis legend was added to the Figures in order to avoid misinterpret. 4. This result surprised us! Figure 5 shows the solar velocity for each one of the HILDCAA intervals, where intervals H03 and H04 are representative for the Autumn season in the South hemisphere (Table 1). Figure 4 also clearly shows the 24-hour oscillation (top left panel). The behavior is very similar to the daily oscillation, where the maximum density occurs during the daytime, while the minimum density occurs at night. However, as has been said, this behavior is already subtracted from the quiet day pattern. Our hypothesis is that because geomagnetic activity during HILDCAAs events remains high, continuous injection of energetic particles can cause prolonged changes in the global wind circulation, causing the ionospheric F layer to remain elevated during the day causing increases in ionization. However, we would like to emphasize that only two events occurring in each season were chosen for this study, and therefore further studies on this specific behavior of dTEC during HILDCAA intervals will need to be performed.

Responses to minor comments: 1. We would like to thank the Reviewer for this note. The words were corrected in the manuscript. 2. We would like to thank the Referee for this suggestion. The word was changed to the manuscript. Please, check the line 67. 3. We would like to thank the Referee for this observation. The sentence was corrected. Please, see the line 97. 3. We would like to thank the Referee for this correction. 4. The sentence was corrected changing "to" to "for". Please, verify the lines 154 - 155. 5. Yes, we used the same scale range for all HILDCAA intervals observed in each GNSS station. We seek the maximum and minimum values to use as a pattern to analyze the hourly distribution. Doing this way, it allows us to compare events with each other. Following the Referee's suggestion, the below sentences were added to the paper (lines 155 - 158): "These values were considered to perform the TEC hourly distribution, i. e., for each specific GNSS station, the maximum and minimum TEC values were used to analyze all HILDCAAs in the same range." 6. We would like to thank the Referee for this observation. 7. We would like to thank the Referee for this observation. The sentence was correct in the manuscript. Finally, we would like to take this opportunity to thank the Reviewer for his/her contribution to improving this work.

Please also note the supplement to this comment:
https://www.ann-geophys-discuss.net/angeo-2019-105/angeo-2019-105-AC2-supplement.pdf

[Figure]

**Supplement:**

[revised manuscript text omitted]

---

## Author Response (AR1)

**REPLY TO EDITOR**

Reply: The authors reviewed the text according to the suggestions and comments proposed by the Referees. In addition, we highlight the modifications in the manuscript to be followed in the review process. Lastly, we would like to take this opportunity to thank the editor for considering this article for publication in Annales Geophysicae.

**REPLY TO REFEREE#1 (Thana Yeeram)**

**Reviewer comments for authors:**

I have read the manuscript that reports the responses of TEC to HILDCAA intervals over the two Brazilian GNSS stations. The manuscript is fairly good in presentation, particularly in reporting an equinoctial anomaly of the TEC during HILDCAAs. As known, this topic has not been extensively studied and is in progress in the field. Since some of issues in the present forms are not adequately explained for the underlying Physics, I decided a minor revision for this manuscript.

Reply: We would like to thank the useful suggestions and comments given by the Referee to improve this paper.

In what follows, we describe how we revised the manuscript and provided the answers to the specific and technical comments of the Referee. We try to make the manuscript clearer and more consistent with what was actually done in this work. We would like to take this opportunity to thank the Referee for his contribution to improving this work.

**Specific comment:**

1. The abstract should be written in a concise form for the lines 32 - 39.

Reply: We would like to thank the Reviewer for this suggestion. We modified the abstract in order to let it more concise.

2. I think that one of the HILDCAA's criteria is there are HSS and high frequency fluctuations of IMF Bz about zero value.

Reply: Actually, the criteria of the fluctuation of IMF Bz around zero was added in a later study carried out by Koga et al. 2011 (doi: 10.1016/j.jastp.2010.09.002). According to Tsurutani et al., 2004 (doi: 10.1016/j.jastp.2003.08.015) the same physical process may occur weather one criteria is not strictly followed.

3. In lines 65 - 67, the authors may refer to n° 2 for HILDCAAs' properties as well.

Reply: We would like to thank the Referee for this suggestion. However, as the information written in the lines 65-67 is well-known, we have preferred to let it in this way.

4. In lines 74 - 75, the authors should mention the references.

Reply: Thank you for the suggestion. We added the reference.

5. In section Data and Methodology, the authors should remove the links of data since they are already in the acknowledgement.

Reply: Thanks for your suggestion. We removed the link of data from the Data and Methodology section and left it in the Acknowledgement section.

6. In line 218 please give more details about the related mechanism for the equinoctial anomaly.

Reply: Information about the equinoctial anomaly is already written in the manuscript in lines 223 - 225. For more details, please see Balan et al., 1998 (doi:10.1029/97JA03137), Mansilla et al., 2005 (doi:10.1016/j.jastp.2005.02.024) and Chen et al., 2012 (doi:10.5194/angeo-30-613-2012).

7. In section 3.3 why the solar wind speed is thought to be a main factor that affects the TEC?

Reply: We would like to thank the referee for this question. We realize that several papers address the relation between HILDCAAs and HSS/CIRs. However, how this relation is done is still an open question. Contrary to our expectations, the TEC answers during HILDCAAs are no direct relation to fast speeds.

- 8. I would like to suggest some related work that may fulfill the discussion: For recent study of TEC and HILDCAAA:
- de Siqueira Negreti, P. M., de Paula, E. R., and Candido, C. M. N.: Total electron content responses to HILDCAAs and geomagnetic storms over South America, Ann. Geophys., 35, 1309–1326, https://doi.org/10.5194/angeo-35-1309-2017, 2017.
  For PPEF and DDEF during HILDCAA:
- Yeeram, T. (2019). The solar wind magnetospheric coupling and daytime disturbance electric fields in equatorial ionosphere during consecutive recurrent geomagnetic storms, Journal of Atmospheric and Solar-Terrestrial Physics, 187, 40-52.
- Yeeram, T., and Paratrasri, A. (2018). Recurrent geomagnetic storms and equinoctial ionospheric F-region in the low magnetic latitude: a case study, IOP Conf. Series: Journal of Physics: Conf. Series 1144: 012024(1-4).

Reply: We would like to thank the Referee for the suggestions. We read them and added the citation in lines 75, 95 and 237.

9. Line 251, the geoeffectiveness of HILDCAA can be different and separated from CIR. Please see Hajra et al. 2015. Relativistic electron acceleration during HILDCAA events: are precursor CIR magnetic storms important?

Reply: Yes, the geoeffectiveness of HILDCAAs can be separated from CIRs, since to a lesser extent; there are HILDCAA events related to CME.

10. The conclusions should be written in a short and concise form.

Reply: We would like to thank the suggestion. The main results were written in topics in the Conclusions section.

11. I do not understand the x-axis of Figures 2 - 5. Why the scale is for one day and for what? I see HILDCAA intervals are longer than that for each event H? Please describe in the text.

Reply: The scale is not for one single day. The x-axis in Figures 2, 3 and 4 represent the mean dTEC hourly values. For Figures 2 and 3, the mean values were represented in this way in order to allow the comparison between one interval and other (please, see the lines 147 - 148). For Figure 4 the x-axis represents the central tendency for all dTEC values, minute-to-minute, as can be seen in lines 215 - 218.

Lastly, in Figure 5, the x-axis represents the time duration of all intervals, represented each 12 hours.

**The technical comments:**

 Some of the English must be corrected. For example, line 251, gerund "Summarizing" should be →In summary/In conclusion.

Reply: We would like to thank the Referee for this correction. The word was changed in the manuscript.

2. The sentences must be simple and concise.

Reply: We are very thankful to the Referee to help us improve the manuscript.

**REPLY TO REFEREE#2**

**Reviewer comments for authors:**

This work deals with a topical issue, which is space weather and its effect on the Ionosphere. However I cannot see which is the real message. If I am right, I understand that the authors want to show that HILDCAAs affect TEC at low and equatorial latitudes. But in my opinion, they need to show that what they find is not something at random, and to exclude any other source of disturbance. I think that the paper may become acceptable for publication after some revision.

Reply: We would like to thank useful suggestions and comments given by the Referee to improve this paper.

In what follows, we describe how we revised the manuscript and provided the answers to the main and minor comments of the Referee. We try to make the manuscript clearer and more consistent with what was actually done in this work. Again, we would like to take this opportunity to thank the Referee for his/her contribution to improving this work.

**Main comments:**

1. I would add some statistical analysis, or qualitative analyses, to show that the TEC disturbances they observe are due to HILDCAAs; by showing that variation they see is not at random and is not due to any other mechanism. Is it possible that it is due to geomagnetic storms, for example? Maybe I am not fully understanding HILDCAAs.

Reply: Unfortunately, do a statistical analysis with ten HILDCAA events as a sample is not significant. However, we believe that carried out a qualitative analysis when we seek to comprise the HILDCAAs disturbances regards seasonal behavior or local time, for example. All TEC disturbances showed in the study is about HILDCAAs influences, since all analysis was done taking into account the dTEC (dTEC = TEC mean – TEC quiet days). Please, see lines 125 - 128.

HILDCAAs are manifestations of space weather events in the form of geomagnetic activities. Their main feature is a continuous flux of energetic particles into the magnetosphere. The average Dst remain suppressed for days or weeks, and appear as an

unnaturally long storm recovery phase. For more, please see Tsurutani and Gonzalez, 1987 (doi: 10.1016/0032-0633(87)90097-3), Tsurutani et al., 2004 (doi: 10.1016/j.jastp.2003.08.015), Hajra et al., 2013 (doi: 10.1002/jgra.50530) and Silva et al., 2017 (doi: 10.5194/angeo-35-1165-2017).

2. Line 165: "The HILDCAA intervals present the positive dTEC predominance" can you quantify this?

Reply: We would like to thank the Referee for this suggestion. We realized that the sentence was short and no concise. To improve the understanding, we have adding the following sentences (lines 168 - 170):

"60% (70%) of all intervals present a positive dTEC response during the whole event for São Luís (Cachoeira Paulista)."

The percentage is regarding the predominance of the positive dTEC during the whole hourly distribution. Only four events for São Luís (three for Cachoeira Paulista) had negative predominance of the TEC behavior.

3. In Figures 2 and 3: what is the "y" axis?

Reply: Figures 2 and 3 are the subplots of all HILDCAA intervals regarding mean dTEC hourly distribution, i. e., each panel refer to each interval. The y-axis is the HILDCAA interval according to its identification (H01, H02, etc).

However, the y-axis legend was added to the manuscript in order to avoid misinterpret.

4. In Figure 5, for H03 and H04 a clear daily variation can be noticed. This 24-hour oscillation is absent in the others. What does this mean?

Reply: This result surprised us! Figure 5 shows the solar velocity for each one of the HILDCAA intervals, where intervals H03 and H04 are representative for the autumn season in the South hemisphere (Table 1). Figure 4 also clearly shows the 24-hour oscillation (top left panel).

The behavior is very similar to the daily oscillation, where the maximum density occurs during the daytime, while the minimum density occurs at night. However, as has been said, this behavior is already subtracted from the quiet day pattern. Our hypothesis is that because geomagnetic activity during HILDCAAs events remains high, continuous injection of energetic particles can cause prolonged changes in the global wind circulation, causing the ionospheric F layer to remain elevated during the day causing increases in ionization.

However, we would like to emphasize that only two events occurring in each season were chosen for this study, and therefore further studies on this specific behavior of dTEC during HILDCAA intervals will need to be performed.

**Minor comments:**

 Line 53: "The", should be lower-case, that is "the", and also in the following "The" which appear after a semi colon.

Reply: We would like to thank the Reviewer for this note. The words were corrected in the manuscript.

2. Line 67: I thinking that may be "differential" should be "different".

Reply: We would like to thank the Reviewer for this suggestion. The word was changed to the manuscript. Please, check the line 67.

3. Line 97: "taking account" should be "taking into account".

Reply: We would like to thank the Referee for this observation. The sentence was corrected. Please, see the line 97.

3. Line 118: use dot for decimal of latitude and longitude instead comma. That is "2,59" should be 2.59.

Reply: We would like to thank the Referee for this correction.

4. Line 153: Why "to São Luís". Shouldn't it be "for São Luís" or at São Luís? The same in the case of Cachoeira Paulista.

Reply: The sentence was corrected changing "to" to "for". Please, verify the lines 154 - 155.

5. Line 154: What do you means by "It was considered the same minimum and maximum values occurred to all intervals, for each station." That you used the same scale range? If this is the case, then you should explain it properly.

Reply: Yes, we used the same scale range for all HILDCAA intervals observed in each GNSS station. We seek the maximum and minimum values to use as a pattern to analyze the hourly distribution. Doing this way, it allows us to compare events with each other.

Following the Referee's suggestion, the below sentences were added to the paper (lines 155 - 158):

"These values were considered to perform the TEC hourly distribution, i. e., for each specific GNSS station, the maximum and minimum TEC values were used to analyze all HILDCAAs in the same range."

6. Line 162: "...phases that over equatorial latitudes", may be you mean "...phases over equatorial latitudes"?

Reply: We would like to thank the Referee for this observation.

7. Line 228: "There are considerable works whose show" should be "There are considerable works that show..."

Reply: We would like to thank the Referee for this observation. The sentence was corrected in the manuscript.

[revised manuscript text omitted]